# Combined Turmeric, Vitamin C, and Vitamin D Ready-to-Drink Supplements Reduce Upper Respiratory Illness Symptoms and Gastrointestinal Discomfort in Elite Male Football Players

**DOI:** 10.3390/nu16020243

**Published:** 2024-01-12

**Authors:** David J. Clayton, Ross Burbeary, Connor Parker, Ruth M. James, Chris Saward, Eleanor L. Procter, William J. A. Mode, Carla Baker, John Hough, Neil C. Williams, Harry Rossington, Ian Varley

**Affiliations:** 1Musculoskeletal Research Group, School of Science and Technology, Nottingham Trent University, Nottingham NG11 8NS, UK; connor.parker@ntu.ac.uk (C.P.); ruth.james@ntu.ac.uk (R.M.J.); chris.saward@ntu.ac.uk (C.S.); ellie.procter2023@my.ntu.ac.uk (E.L.P.); william.mode2015@my.ntu.ac.uk (W.J.A.M.); carla.baker@ntu.ac.uk (C.B.); john.hough@ntu.ac.uk (J.H.); neil.williams@ntu.ac.uk (N.C.W.); ian.varley@ntu.ac.uk (I.V.); 2Derby County Football Club, Pride Park, Derby DE24 8XL, UK; ross.burbeary@dcfc.co.uk; 3Lincoln City Football Club, LNER Stadium, Lincoln LN5 8LD, UK; hros@theredimps.com

**Keywords:** curcumin, soccer, recovery, inflammation, gut permeability, URTI, wellness

## Abstract

Elite football is associated with the increased risk of illness, although targeted supplementation can reduce illness risk. This study assessed the effects of a supplement containing turmeric root within a black pepper and fat-soluble blend, vitamin C and vitamin D, on upper respiratory symptoms (URS), gastrointestinal symptoms (GIS), muscle soreness, and markers of inflammation and gut permeability in elite male footballers. Twenty-three footballers completed 3 weeks of no intervention (CON), followed by 16 weeks of daily consuming 60 mL of a commercially available supplement containing raw turmeric root (17.5 g, estimated to contain 700 mg of curcumin), vitamin C (1000 mg), and vitamin D3 (3000 IU/75 mcg) (SUP). URS and GIS were measured daily. Immediately (0 h), 40, and 64 h after six competitive matches (two in CON, four in SUP), the subjective soreness and plasma concentrations of creatine kinase [CK], c-reactive protein [CRP], and intestinal fatty-acid binding protein [I-FABP] were assessed. URS incidence (*p* < 0.001), GIS (*p* < 0.05), and plasma [I-FABP] at 0 h (*p* < 0.05) were greater during CON versus SUP. At 40 h, [CRP] was greater than 0 h during CON (*p* < 0.01) but not SUP (*p* = 0.204). There were no differences in soreness or [CK]. This study indicates that turmeric root, vitamin C, and vitamin D supplementation over 16 weeks can reduce URS, GIS, and post-match [I-FABP] in elite footballers.

## 1. Introduction

Illness and injury have been shown to substantially limit player availability for both training and competition in elite sport [1,2]. In team sports, including football, significant relationships have been evidenced among increased training loads, upper respiratory illnesses (URI), and decreases in a primary antibody in saliva, IgA [3,4,5]. It is also known that the lowering of IgA in saliva is positively associated with increases in URI [6]. In addition to a decline in salivary IgA, a post-exercise decline in immunosurveillance has been reported with prolonged (>5 days) and intensive (>60% VO_2max_) training periods [7], and alongside a post-exercise decline in cytotoxic T cells may introduce a window of opportunity for infection [8]. In professional football, it has been demonstrated that illness accounts for 6.3% and 8.5% of absences in training and match-play, respectively [1]. Football teams with less player absences achieve a higher league ranking and a greater number of points per match [9], hence illnesses could have a direct effect on a team’s success. 

The intensity and training demands of professional football cause substantial muscle damage and inflammation, particularly after match-play [10]. The limitations of the antioxidant system to neutralize free radicals delays recovery and causes sustained muscle soreness referred to as delayed onset muscle soreness (DOMS) [11]. In addition, the delayed regulation of free radicals can negatively impact the operation of the immune system [12,13]. Professional footballers are often required to play several matches in close proximity (48–72 h between matches), which curtails recovery time and may increase the risk of illness [14]. Professional footballers often rely on non-steroidal anti-inflammatory drugs (NSAIDs) to aid recovery [15]. However, the long-term use of NSAIDs is associated with several adverse side-effects [16], necessitating the exploration of more effective and safer solutions. 

An alternative to NSAIDs may be supplementation, which can enhance the availability of certain nutrients or increase the antioxidant capacity, which may reduce inflammation and discomfort [11]. Micronutrient deficiency can increase illness risk, and the current guidance states that athletes should consume vitamin C and vitamin D supplements during periods of high exertion to reduce the incidence and severity of upper respiratory tract symptoms [11]. Vitamin D holds an important role in our immunity and in the modulation of inflammation [17] with almost all immune cells expressing vitamin D receptors [18]. Vitamin C is an antioxidant that accumulates in leukocytes and can protect cells from oxidative damage [19]. Vitamin C can regulate immune function by modulating redox-related cell signaling pathways or by directly protecting important cell structural components. There is some indication that vitamin C may support the ability of neutrophils to migrate towards a site of infection and enhance the chemotaxis capability of neutrophils. 

There is also emerging evidence that anti-inflammatory supplements can reduce DOMS latency and restore muscle function more rapidly after muscle damaging exercises [11]. Curcumin, found in turmeric, has been shown to reduce inflammation and muscle soreness and accelerate the restoration of muscle function [20,21,22]. Supplementing professional footballers with 70 g of turmeric root per day (approximately 1400 mg of curcumin) can reduce the markers of inflammation and perceived muscle soreness after match-play [23]. There is also some evidence that curcumin, vitamin D, and vitamin C can all improve gut barrier functioning and influence gut microbiota [24,25,26]. This suggests an interesting mechanism of action that requires further investigation. 

Gastrointestinal discomfort is a common issue reported by elite athletes [27], which can have serious implications on health and performance. Although the exact mechanisms of exercise-induced gastrointestinal symptoms (GIS) are not fully understood, increased splanchnic hypoperfusion and intestinal ischemia are considered key contributors [28]. Submaximal exercise reduces splanchnic blood flow by 80% [29], altering intestinal barrier functioning and increasing gut permeability [30]. This can lead to increased GIS, systemic inflammation, and nutrient malabsorption [31], which can have detrimental effects on performance and recovery. 

The intensity, training demands, and match-play regularity of professional football considerably increases the propensity for illness and injury. Nutritional aids that can support better immune function and expedite post-match recovery could increase player availability and improve performance. The aim of this study is to assess whether a combined turmeric, vitamin C, and vitamin D supplement could reduce illness prevalence, gut permeability, markers of inflammation, and symptoms of DOMS in professional footballers following match-play. 

## 2. Methods

### 2.1. Participants

Twenty-three elite male (age: 24.3 ± 2.8 year, weight: 79.7 ± 6.7 kg, height: 1.83 ± 0.06 m) professional footballers (Tier 4; [32]), competing in the English third tier during the 2022/23 season, volunteered to participate in this study. To qualify for enrolment, participants were required to be outfield players with no history of cardiovascular or gastrointestinal complaints and not regularly consuming any vitamin C, D, or turmeric supplements. All participants provided written consent after they were informed verbally and in writing of the nature and requirements of the study. The study was approved by the Nottingham Trent University Human Invasive Ethics Committee (REF: 716).

### 2.2. Study Design and Procedures

This study was a non-randomized within group intervention, taking place during the competitive season. Participants completed a 22 day (totaling 484 player days) control period during which no supplement was consumed (CON), followed by a 116 ± 8-day (totaling 2558 player days) intervention period where they consumed a commercially available 60 mL “shot” each day (Immunity Support Vitamin C + D, The Turmeric Co., Cambridgeshire, UK) (SUP). The shot was typically consumed in the morning, post breakfast and prior to training. Each shot contained 17.5 g of raw turmeric root (estimated to contain 700 mg curcumin), 1000 mg of Vitamin C, 3000 IU of Vitamin D3, and 200 mg of black pepper (estimated to contain 10 mg of piperine). Compliance with supplementation was verbally confirmed by participants prior to each match. Participants also reported the incidence of upper respiratory symptoms (URS) and gastrointestinal (GI) distress via a daily questionnaire (Figure 1). 

CON took place in September 2022 and comprised two competitive league matches, while SUP took place between November 2022 and February 2023 and comprised four competitive league matches. Immediately, 40 h, and 64 h after each match, subjective soreness and a capillary blood sample were collected. Global positioning system (GPS, Vector, Catapult, Australia) was used to monitor common physical performance measures (total distance, high speed distance, accelerations, deceleration) throughout the control and intervention periods during match-play. 

### 2.3. Illness and Wellbeing Measurements

To assess the presence of upper respiratory symptoms, participants completed the Jackson scale questionnaire, which has previously been validated against known infections [33]. Participants were instructed to rate the presence of eight symptoms (headache, chilliness, sneezing, sore throat, malaise, coughing, nasal discharge, and nasal obstruction) over the previous 24 h on a scale of 0 to 3 (0—not present, 1—mild, 2—moderate, 3—severe). To assess GI distress, participants used a standardized questionnaire previously demonstrated to detect changes in GI symptoms in athletes [34], whereby participants rated the presence of global GI symptoms using a subjective analogue scale ranging from 0 to 9 (0—“no problem at all”, 9—“worst it’s ever been”). Participants completed these scales each day on their smart phones.

### 2.4. Blood Collection and Analysis

Capillary blood samples (300 μL) were collected in EDTA collection tubes (Microvette, Sarstedt, Germany). The post-match samples were collected where the match took place, with follow-up samples collected on arrival at the training facility. Samples were stored on ice after collection (maximum of 30 min) before being centrifuged (13,000× *g*, 10 min, 4°C), the plasma aliquoted, and storage at −80 °C. All plasma samples were analyzed for [CK] and [CRP] using an ABX Pentra 400 (Horiba Medical, Kyoto, Japan; [CK] CV: 1.7%; [CRP] CV: 0.6%). Baseline and post-match plasma samples were also analyzed for [I-FABP] via ELISA (Hycult Biotechnology, Uden, The Netherlands; CV: 1.7%).

### 2.5. Subjective Soreness Measurements

Subjective whole-body and leg-specific soreness were assessed using paper-based, valid, reproducible [35], 100 mm visual analogue scales, anchored at 0 mm with ”no pain” and at 100 mm with “as much pain as it could be”. Participants were asked to apply a mark on the line to indicate their level of soreness, which was then quantified with a ruler. Subjective soreness measures were taken immediately (0 h), 40 h, and 64 h after each match. 

### 2.6. GPS

The physical demands of match-play and training were monitored using a 10 Hz GPS (S7, Vector, Catapult, Victoria, Australia). This system has been valid and reliable to assess athlete movements [36]. Each player wore a GPS unit positioned between the shoulder blades using a bespoke garment. Each GPS unit was downloaded after each match or training session, and analyzed using commercially available software ( Open field, Catapult, Victoria, Australia). The physical performance variables assessed included the following: total distance covered (m), high speed (>5.5 m/s) distance covered (m), very high speed (>7.0 m/s) distance covered (m), number of accelerations above 0.5 m/s^2^ for >0.5 s, and number of decelerations below −0.5 m/s^2^ for >0.5 s.

### 2.7. Statistical Analysis

A Pearson chi-square test (χ^2^) was used to assess the observed frequency of URS in CON and SUP. Paired sample *t*-tests were conducted to assess differences in GI symptoms relative to the days in each trial. A subset of participants who competed at least one match in both CON and SUP were assessed for subjective markers of soreness (leg-specific and whole body) (*n* = 8), markers of inflammation ([CK] and [CRP]) (*n* = 8), and [I-FABP] (*n* = 7). An average value for CON and an average value for SUP were calculated for players that played in more than one match during the control or supplementation periods, respectively. Soreness and inflammation markers (*n* = 8) were analyzed via a two-way repeated measures analysis of variance (ANOVA). 

The significant main effects were followed-up using Bonferroni-corrected paired *t*-tests. For [I-FABP], the average post-match and baseline-post match absolute and change from baseline values were calculated during the control and supplementation periods (*n* = 7). Differences between the trials were assessed using a paired sample *t*-test. A Paired samples *t*-test were conducted to assess the differences in GPS metrics between the control and the intervention periods. The statistical significance was accepted at the 95% confidence level (*p* < 0.05). The mean and standard error were used to describe the average and variability of data, unless stated otherwise. The effect sizes were calculated for chi-squared tests using Cramer’s V (small = 0.10–0.29, medium = 0.30–0.49, large ≥ 0.50), for *t*-tests using Cohen’s dz (small = 0.20–0.49, medium = 0.50–0.79, large ≥ 0.80), and for ANOVA using partial eta squared (small = 0.01–0.05, medium = 0.06–0.13, and large ≥ 0.14) [37,38].

## 3. Results

### 3.1. Illness Incidence and Gastrointestinal Distress

There was greater incidence of URS during CON compared to SUP (Chi Sq = χ^2^ 75.7, DF = 1, *p* < 0.001, V = 0.16). Illness incidence was reported as 3.9 per 1000 player days during SUP compared to 10.6 per 1000 player days during CON (Figure 2). This equated to 94.2% of time spent without a URS during SUP compared to 83% of time spent without a URS during CON. 

Relative GI distress was significantly lower during SUP (43 GI discomfort score per 1000 player days) compared to the CON (111 GI discomfort score per 1000 player days) period (*p* < 0.05, dz = 0.62).

### 3.2. Markers of Gut Damage

The average post-match [I-FABP] was lower during SUP than CON (SUP: 735 ± 275 pg/mL; CON: 1821 ± 1280 pg/mL; *p* < 0.05, dz = 0.98) (Figure 3).

### 3.3. Markers of Inflammation

There was no effect of trials (*p* = 0.635, η^2^p = 0.03), but there were main effects of time (*p* < 0.01, η^2^p = 0.56) and a time-by-trial interaction (*p* < 0.001, η^2^p = 0.65) effect for plasma [CRP]. There were no significant changes in [CRP] between time points during SUP (*p* ≥ 0.204, dz = 0.23–0.42), whereas [CRP] increased between 0 h and 40 h (*p* < 0.01, dz = 1.58) and decreased between 40 h and 64 h (*p* < 0.01, dz = 1.76) during CON. There were no differences in [CRP] between trials at any timepoint after correction for multiple comparisons (*p* ≥ 0.258). There were no main time (*p* = 0.059, η^2^p = 0.40), trial (*p* = 0.831, η^2^p = 0.01) or time-by-trial interaction (*p* = 0.171, η^2^p = 0.25) effects for plasma [CK] (Figure 4).

### 3.4. Subjective Markers of Soreness

There was a main effect of time (*p* < 0.001, η^2^p = 0.92) but no trial (*p* < 0.892, η^2^p = 0.003) or time-by-trial interactions (*p* < 0.447, η^2^p = 0.11) main effects for leg soreness. Leg soreness decreased progressively over time. Leg soreness was lower at 40 h than 0 h (*p* < 0.001, dz = 1.16) and lower at 64 h than 40 h (*p* < 0.001, dz = 1.14) in CON and lower at 40 h than 0 h (*p* < 0.001, dz = 1.90) and lower at 64 h than 40 h (*p* < 0.001, dz = 1.85) in SUP (Figure 4). There was a main effect of time (*p* < 0.001, η^2^p = 0.86) but no trial (*p* = 0.968, η^2^p < 0.001) or time-by-trial interaction (*p* = 0.944, η^2^p = 0.12) main effects for whole-body soreness. Similar to leg soreness, whole body soreness decreased progressively over time. Whole-body soreness was lower at 40 h than 0 h (*p* < 0.01, dz = 0.83) and lower at 64 h than 40 h (*p* < 0.001, dz = 4.3) in CON and lower at 40 h than 0 h (*p* < 0.01, dz = 1.51) and lower at 64 h than 40 h (*p* < 0.001, dz = 2.15) in SUP (Figure 5).

### 3.5. GPS Data

There was no difference between trials for total distance covered (CON: 9845 ± 1026 m per match; SUP: 9883 ± 437 m per match; *p* = 0.95), high speed distance covered (CON: 633 ± 76 m per match; SUP: 534 ± 90 m per match; *p* = 0.19), number of accelerations above 0.5 m/s^2^ for >0.5 s (CON: 28 ± 3 *n* per match; SUP: 26 ± 2 *n* per match; *p* = 0.40), or number of decelerations below −0.5 m/s^2^ for >0.5 s (CON: 39 ± 5 *n* per match; SUP: 41 ± 4 *n* per match; *p* = 0.56). 

## 4. Discussion

The primary finding from this study was that a supplement containing 17.5 g of raw turmeric root (~700 mg curcumin), 1000 mg of Vitamin C, 3000 IU of Vitamin D3, and 200 mg of black pepper (~10 mg of piperine) reduced the incidence of upper respiratory and GI symptoms. These findings suggest that this combination supplement, consumed once a day, could be an effective method of reducing illness symptoms in elite male footballers. 

Illness incidences in the present study were reported as 3.9 (SUP) and 10.6 (CON) per 1000 player days. Previously, illness has been shown to occur in 7–17% athletes competing at major sporting events [39] and has an incidence of 0.8 illnesses per 365 athlete days [40]. In professional football, illnesses have an incidence of 0.6 and account for 2.5 days lost per 1000 player hours [41]. The incidence of illness in the present study is greater than what has previously been reported, however the contagious nature of illness could explain these findings. While illnesses have less influence on player availability compared to injury, the knowledge that illness can limit performance [42] makes illness mitigation an important aim for elite sport practitioners.

The use of a novel combined (vitamin C, vitamin D and turmeric root) supplement in the present study may target various areas of illness manifestation. There are some mechanistic links between vitamin D and the physical barrier in the respiratory system; for example, it supports the production of proteins of tight junctions leading to an improved barrier to reduce the risk of infection [43]. Cross-sectional studies have reported that maintaining sufficient vitamin concentrations is associated with a reduction in respiratory illnesses in both military personnel and elite athlete groups. Specifically, military personnel with insufficient vitamin D3 are nearly two times more likely to miss duty due to respiratory infections [44]. In addition, athletes with positive respiratory infections confirmed via virology/bacteriology analysis have lower vitamin D concentrations compared to those with negative respiratory infection results [45]. It is accepted that these cross-sectional studies do not provide a cause-and-effect conclusion. Vitamin C influences a number of immune functions; for example, high concentrations of vitamin C are associated with enhanced antibody responses [46]. Curcumin is known to alter the COX-2 pathway signaling leading to the lowering of a number of inflammatory cytokines [47]. Tanabe et al. [48] showed that the ingestion of 180 mg/d of curcumin for 7 days led to an attenuation of the inflammatory cytokine IL-8 response in the 12 h following maximal isokinetic exercises compared to a placebo group. Due to the combined nature of the supplement used in the present study, it is impossible to confirm the dominant mechanism/pathway response for lowering illness incidence.

This study found that the supplement reduced I-FABP, a marker of intestinal damage. I-FABP has consistently been shown to increase post-exercise, suggesting that high-intensity or long-duration exercise can have an acute damaging effect on the gut barrier, which may exacerbate inflammation and recovery [49]. The current study is the first to demonstrate that a combined supplement of vitamin D, vitamin C, and turmeric root may have a role in maintaining intestinal barrier integrity and reducing the post-exercise damage to the gut barrier. Vitamin D plays a pivotal role in the maintenance of gastrointestinal barrier integrity by regulating proteins associated with intestinal epithelial gap junctions [50]. The biological activity of vitamin D is mediated by vitamin D receptors that are abundantly expressed in the intestine, which suggests a role of vitamin D status in gut barrier integrity. Curcumin, one of the active ingredients of turmeric root, has also been shown to reduce post-exercise increases in I-FABP compared to a placebo after just 3 days (500 mg/day) [51]. Furthermore, reduced gut damage may also be linked to vitamin D’s and turmeric’s purported roles in maintaining gut microbiota homeostasis. The gut microbiota directly interacts with epithelial cells and the mucosal lining of the gut, which acts as a physical barrier and regulates translocation of pathogens and toxins into circulation. Therefore, the reduced gut damage following exercise because of the combined supplement may also support the tendency to see reductions in pro-inflammatory mediators in the current study.

Plasma [CRP] increased 40 h post-match, but the increase was attenuated in the supplementation period compared to the control period. This provides tentative support to our previous study that found a turmeric-containing supplement (70 g/d) reduced [CRP] in elite male footballers compared to a non-supplementation group [23]. CRP is a protein secreted by the liver that correlates with systemic inflammation [52]. Curcumin has been previously shown to downregulate inflammatory regulators and increase antioxidant capacity [53,54], which may explain why the [CRP] response was attenuated during the supplementation period. Despite within-trial [CRP] differences after match-play, there were no differences between trials after correction for multiple comparisons. This could be due to the turmeric dose (17.5 g/d), estimated to provide ~700 mg of curcumin, which was lower than the 70 g/d (~2800 mg/d curcumin) dose previously shown to reduce plasma [CRP] in elite footballers [23]. However, the supplement provided also contained vitamin C, which has anti-inflammatory and antioxidant properties [55]. As such, the addition of vitamin C to the low-dose turmeric may have contributed to the within-trial attenuation of [CRP] observed in this study, but the effect may be greater at a higher turmeric dose.

There were no differences in plasma [CK] within or between trials, which reflects a previous turmeric supplementation study on elite footballers [23]. CK is a biomarker protein for muscle damage [56]. However, CK does exhibit large inter-person variability [57], and the CK response to exercise is influenced by training status and muscle fiber type [58,59,60]. The applied nature of this study in an elite population prevented tight control over participants’ diets and exercise patterns, which may have influenced the responses.

The subjective markers of soreness decreased progressively after match-play, but there was no difference between the control and supplementation periods. Whilst these findings contrast some studies that have observed a reduction in subjective soreness after turmeric or curcumin supplementation [20,23,48], there are studies that report no subjective differences after curcumin supplementation [22,61,62], which in some instances contrast with changes in biomarkers of inflammation [22,61]. It is evident from previous studies that continuing curcumin supplementation during the recovery period is important to achieve analgesic effects. Tanabe et al., [48] found that supplementing 180 mg/d of curcumin for 4 days after eccentric exercise reduced subjective soreness, whereas 180 mg/d for 7 days before exercise had no effect on soreness. In the current study, daily supplementation persisted through the pre- and post-exercise period. The curcumin dose was lower than a previous study [23] but similar to [20], or greater than [48], other studies that have reported a positive effect on subjective soreness after exercise. Contextual and external factors can inevitably influence a subjective perception of soreness, and these factors are difficult to control in elite sportspeople. Coupled with the small sample size included in this secondary outcome (*n* = 8), this may explain the null subjective findings, despite a positive [CRP] change. A fundamental issue in comparing existing literature on curcumin and turmeric supplementation is that few studies conduct pharmacokinetic analysis to determine whether the consumed dose reaches peripheral circulation. Unformulated curcumin is known to have poor bioavailability [53], and whilst there is evidence that bioavailability can be enhanced by formulating curcumin with adjuncts, such as piperine [63], this has rarely been empirically tested. The current study investigated a supplement containing turmeric root (estimated to provide approximately 700 mg of curcumin) and formulated with 10 mg of piperine to enhance bioavailability [63]. However, in accordance with the majority of curcumin and turmeric supplementation studies, curcumin’s appearance in blood was not assessed. It is possible that there were differences in the amount of curcumin that reached peripheral circulation, and how this is dynamically affected by the dosing strategy may explain some of the conflicting findings in the literature on subjective soreness and inflammation biomarkers. Future research should assess the pharmacokinetic profile of functional compounds to provide mechanistic insight into biological functions. 

These findings demonstrate the potential efficacy of a combined supplement regimen, administered conveniently as a 60 mL shot once daily, in mitigating illness symptoms, enhancing gastrointestinal health, and reducing inflammation among elite footballers. The findings are particularly pertinent within the context of elite football, given the multifaceted demands intrinsic of the sport. Activities like frequent travel [64], the inherently physical contact nature of the game [65], and prolonged rigorous training [66] have all been linked to an increased susceptibility to illnesses. Traditional vitamin supplementation, taken to mitigate illness, in tablet or medication form has often exhibited low adherence rates [67]. Anecdotally, participants attested to its user-friendly nature, describing it as “easy to consume”, and reported no instances of gastrointestinal discomfort linked to its usage. This collective evidence demonstrates the potential of this novel supplement approach in optimizing the health of elite footballers. It should be noted that the vitamin D and vitamin C within the supplement was relatively high. It has previously been reported that excessive amounts of vitamin D and vitamin C can cause adverse side-effects including hypercalcemia [68] and GI distress [69]. However, no adverse effects were reported in the present study.

This study has several strengths and limitations. This is one the first studies to explore a supplement of this nature in an applied environment using elite footballers. By virtue of this, it was impossible to blind participants to the study intervention, which may have influenced the findings. There is some evidence that the placebo effect can explain 85% of the reduction in cough symptoms [70] and that perceptions of pain can be reduced when participants perceive they have ingested an active ingredient [71]. As such, the lack of blinding could have influenced illness reporting or subjective soreness outcomes, although no differences were observed in the latter. Future studies should aim to incorporate a suitable placebo to mitigate this risk. Similarly, the applied nature of this study limits control around diet and physical activity. For example, we were not able to strictly control what and when players ate, and we were not able to collect a detailed supplementation history for each player. However, the participants were all professional football players, and although there may be differences in diet and physical activity between training, match, and rest days, it is unlikely that significant variations occurred across the study period. We also instructed players to maintain their usual dietary and supplementation strategies over the course of the study. To improve this aspect, it is recommended that future studies conduct supplementation over longer periods, investigate specific dosing strategies, and explore supplement combinations. 

The results of any study should be applied only to populations similar to those involved in the research design [36]. Thus, although research in sub-elite populations can be informative, the findings cannot be extrapolated to professional football environments. However, conducting highly controlled, longitudinal cross-over designed research in elite environments is inherently difficult [72] due to factors including but not limited to inherently small sample sizes and hesitation from coaches and athletes due to the disruption that the ”best practice” research methodology has on training and competition practices. The fact that these data were collected longitudinally on a whole professional football squad across a competitive season is a major strength of the present work. However, due to a range of factors including player selection, injury, and transfer, markers of inflammation, intestinal damage, and subjective soreness were assessed in a smaller subset of players (*n* = 7–8). A larger sample of player matches is warranted in future research, permitting more sophisticated data analysis procedures to be adopted, such as mixed-modelling [72], and allowing post-match and across season changes in outcomes to be better elucidated. 

## 5. Conclusions

In summary, this study found that the daily intake of a turmeric root, vitamin C, and vitamin D combined supplement reduced GI distress and upper respiratory symptoms in a group of elite male footballers. This study also found that a marker of intestinal damage was also reduced during the supplementation period, indicating a potential mechanism of action. There were no differences between the trials for markers of inflammation or subjective muscle soreness, which may be due to the low dose of turmeric provided in this study. These findings suggest that a combined turmeric root, vitamin C, and vitamin D supplement may be an effective way to reduce illness occurrence in elite male footballers. 

## Figures and Tables

**Figure 1 nutrients-16-00243-f001:**
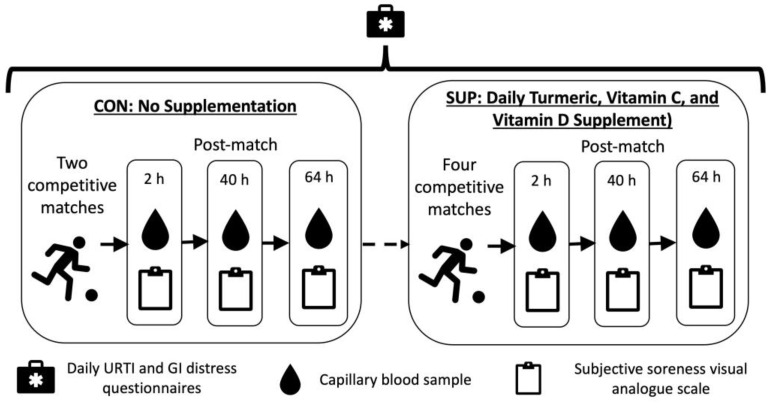
Schematic representation of study design.

**Figure 2 nutrients-16-00243-f002:**
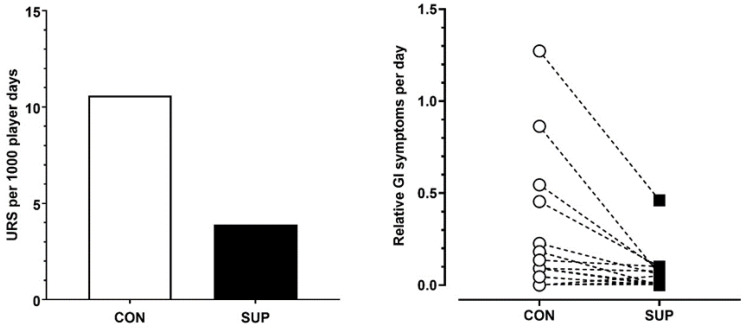
Mean upper respiratory symptoms per 1000 player days (**left panel**) and individual player responses to gastrointestinal symptoms per day (normalized over time) (**right panel**) in the control period (CON) and supplementation period (SUP).

**Figure 3 nutrients-16-00243-f003:**
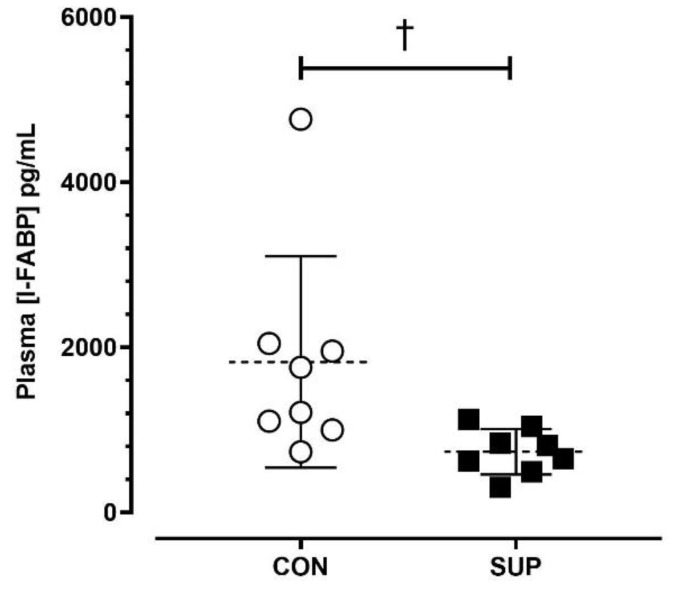
Absolute plasma concentrations of intestinal fatty-acid binding protein ([I-FABP]) immediately post-match in the control period (CON) and supplementation period (SUP). † indicates a difference between trials (*p* < 0.05). Values are means with error bars representing standard error.

**Figure 4 nutrients-16-00243-f004:**
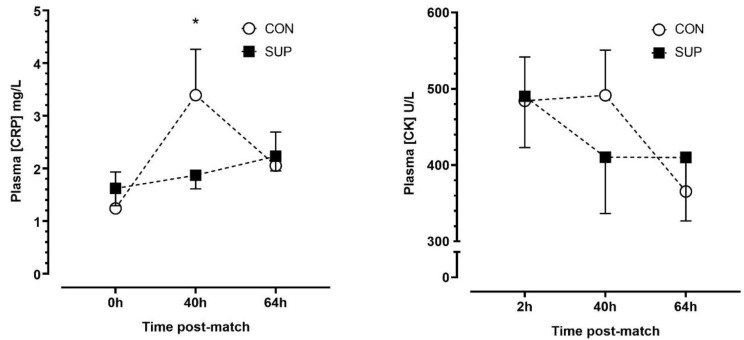
Plasma concentrations of c-reactive protein ([CRP]) (**left panel**) and creatine kinase ([CK]) (**right panel**) immediately (0 h), 40 h and 64 h post-match, in the control period (CON; open circle) and supplementation period (SUP; solid square). * indicates a difference compared to 0 h in CON (*p* < 0.05). Values are means with error bars representing standard error.

**Figure 5 nutrients-16-00243-f005:**
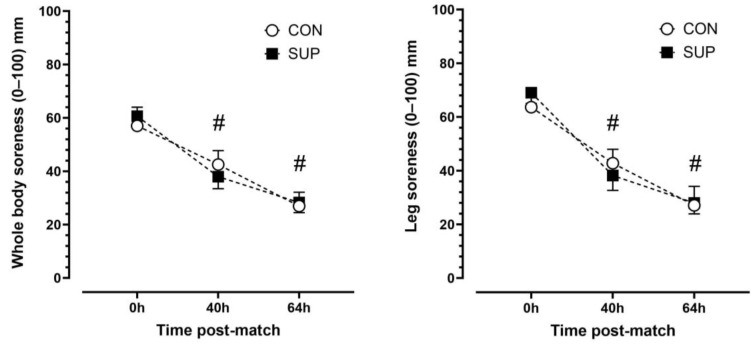
Subjective ratings of leg (**left panel**) and whole-body (**right panel**) soreness immediately (0 h), 40 h, and 64 h post-match in the control period (CON; open circle) and supplementation period (SUP; solid square). # indicates a difference compared to 0 h in both trials (*p* < 0.05). Values are means with error bars representing standard error.

## Data Availability

The data presented in this study are available on request from the corresponding author. The data are not publicly available due to privacy reasons.

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
