# Peer review of "Combined Turmeric, Vitamin C, and Vitamin D Ready-to-Drink Supplements Reduce Upper Respiratory Illness Symptoms and Gastrointestinal Discomfort in Elite Male Football Players"

_nutrients, 2024, doi:10.3390/nu16020243_

Round 1
Reviewer 1 Report
Comments and Suggestions for Authors
I would like to congratulate the authors on developing the study: “Combined turmeric, vitamin C, and vitamin D ready-to-drink food beverage reduces illness symptoms and gastrointestinal discomfort in elite male footballers.
I have some questions to improve the content of this paper.
Insert a flowchart (drop-out rate, split group pieces of information, among others)
Insert a figure with experimental design with details.
Discuss the reproducibility of the instruments used in the present study
Calculate relative and absolute deltas and Cohens d.
Insert the questionnaire and scales used after the references.
Classify eta square using Richardson (2011).
Calculate the sample power of the present study.
Insert the present study’s limitations, strength points, and future directions.
Was the subjective perception or rating of the athletes’ perceived recovery measured?
Wasn't the amount of flavonoids used low? Other foods like coffee also have flavonoids.
Reviewer 2 Report
Comments and Suggestions for Authors
The study provides contribution to the ‘’food first’’ philosophy in sports nutrition, and the next step would be to provide a practical advice for athletes, also tested in a potential following study, more based on whole foods providing comparative benefits relative to commercial product.
1. Title: delete ‘’food’’, is not necessary, beverage is substantial description, what illness? delete, or be specific, football players instead of footballers
2. Abstract: indicate if a beverage is a commercial product, or specially designed for the study; express vitamin D amount in mcg instead of IU, add info for expected curcumin intake
3. Conclusion: add info for a duration of daily supplementation
4. Provide details regarding what time of day, in relation to meals and training, the baverage was taken
5. Comment potential adverse effects, since 16 weeks of 1000 mg vit C and 75 mcg vit D is relatively high dose; add info about taking into account additional intake of vit C and D since cumulative amount could additionally increase the risk of adverse affects
6. Can you add quotes from subjects, regarding feasibility, taste, acceptance, subjective perception, etc. regarding experienced chronic supplementation
7. Please additionally emphasize in ‘’Confliet of Interest’’ (also correct to conflict)’’ that the authors (all), players, club management do/not have a relation with the manufacturer of the product
Reviewer 3 Report
Comments and Suggestions for Authors
This manuscript describes a study which investigates whether a combined turmeric, vitamin C and vitamin D supplement could reduce illness prevalence, gut permeability, markers of inflammation and DOMS in professional football players after a match.
Line 12: Change to "...increased risk of illness..."
The authors state in the abstract and imply throughout the introduction, specifically lines 76 - 79, that the frequency of illness is higher in elite footballers than the general population due to training demands and match-play. How do you know that these illnesses are influenced by training and match-play? Are the rates different from what would be expected in any generally healthy population? Please provide a comprehensive justification for this with references.
The methods and results sections are clearly written and the methods describes a well designed study. The statistical analyses are also appropriate for this data and figures in the results section are helpful for the reader.
The discussion explains the results clearly without overstating the findings.
Round 2
Reviewer 1 Report
Comments and Suggestions for Authors
Dear authors. I would like to congratulate you on the adjustments performed.
However, I have more some questions:
(1) Was an instrument applied to verify the food pattern ingestion of football players?
(2) Was an anamnesis with supplement (ergogenic aids) applied to each participant of the present study?
The questions above should be inserted in limitations if not monitored in the present study.
